# BENCHMARKING FOUNDATION MODELS FOR UNSUPERVISED DISCOVERY IN LARGE MULTIMODAL ASTROPHYSICAL DATASETS

**Maxime Ronceray**
Université de Tours
Laboratoire d'Informatique Fondamentale
et Appliquée de Tours (LIFAT)
Tours, France
maxime.ronceray@etu.univ-tours.fr

**Marc Huertas-Company**
Instituto de Astrofísica de Canarias
Universidad de La Laguna (ULL)
Tenerife, Spain
mhuertas@iac.es

**Alexandre Chanson**
Université de Tours
Laboratoire d'Informatique Fondamentale
et Appliquée de Tours (LIFAT)
Tours, France
alexandre.chanson@univ-tours.fr

**Malgorzata Siudek**
Instituto de Astrofísica de Canarias
Tenerife, Spain
msiudek@iac.es

**Anna Preto**
Université de Paris, CNRS,
Astroparticule et Cosmologie,
F-75013 Paris, France,
Mews Labs, 191 Avenue Aristide Briand,
94230 Cachan, France
anna.preto@apc.in2p3.fr

**Michael J. Smith**
AstroAI
Center for Astrophysics
Harvard & Smithsonian,
UniverseTBD
mike@mjjsmith.com

**Julien Zoubian**
Aix Marseille Univ, CNRS/IN2P3,
CPPM, Marseille, France
zoubian@cppm.in2p3.fr

**Clara Bonini**
Aix Marseille Univ, CNRS,
CNES, LAM, Marseille, France
clara.bonini@lam.fr

## ABSTRACT

We benchmark pretrained multimodal foundation model embeddings for discovery in large scientific datasets through unsupervised anomaly detection. Using a matched imaging–spectroscopy sample from the Euclid and DESI astronomical surveys, we compare three self-supervised representation approaches: autoregressive modeling (AstroPT), contrastive image–spectrum alignment (AstroCLIP), and encoder-decoder multimodal transformers (AION), with only lightweight adaptation. We introduce a scalable pipeline based on density estimation in embedding space and multimodal anomaly isolation, combining per-modality rarity with explicit cross-modal misalignment. A cross-model ranking-transfer analysis shows that the most extreme outliers are often shared across models, while their relative prioritization strongly depends on training objectives and inductive biases, indicating that anomaly definitions are representation-relative. Qualitative inspection suggests that unimodal density tails are frequently dominated by instrumental artefacts, whereas multimodal fusion increases the prevalence of physically coherent candidates such as active galactic nuclei and gravitational lens candidates. Finally, lightweight probing tasks reveal how different embeddings trade predictive accuracy against linear decodability and effective predictive dimensionality. Together, these results provide practical guidance for deploying foundation model embeddings for discovery in upcoming large-scale survey data.

# 1 INTRODUCTION

Rare and unusual objects in astronomical surveys often drive scientific discovery, yet their identification becomes increasingly challenging as survey volumes grow. In large astronomical datasets, statistical outliers may correspond either to rare but physically meaningful systems—such as strong gravitational lenses, active galactic nuclei (AGN), or interacting galaxies—or to instrumental and data processing artefacts (e.g., cosmic rays, saturation effects, diffraction spikes) (Baron & Poznanski, 2017; Margalef-Bentabol et al., 2020). Both categories can appear as outliers under unsupervised criteria; however, deciding which anomalies are scientifically meaningful requires additional information beyond unsupervised rarity alone. This motivates leveraging multimodal structure (e.g., cross-modal agreement/misalignment) rather than relying on unimodal rarity alone. As survey volumes grow rapidly, reliable identification and prioritization of physically meaningful anomalies becomes a key challenge for scientific exploitation (Kurcz et al., 2016; Cheng et al., 2020).

Large astronomical surveys, such as the Euclid Collaboration (et al. Euclid, 2025), provide a particularly demanding testbed for these challenges. In particular, its first public release (Euclid Q1 (Collaboration & et al., 2025)) delivers high-resolution space-based imaging of galaxies. In this work, we use a multimodal dataset of matched Euclid imaging with complementary spectroscopic observations from DESI (Collaboration, 2016). Although the Q1 subset considered here contains on the order of $10^4$ objects, future Euclid data releases will increase this scale by several orders of magnitude, making representation reuse and embedding-based analysis essential for scalable scientific workflows (Baron, 2019; Huertas-Company & Lanusse, 2023; Smith & Geach, 2023).

From a machine learning perspective, the challenge of identifying rare astrophysical anomalies can be framed as an unsupervised anomaly detection problem under limited supervision, extreme class imbalance, and high-dimensional, heterogeneous observations. Recent progress in machine learning has shown that large pretrained models can learn reusable representations that transfer across tasks and domains, enabling scalable analysis without retraining models from scratch (Hinton et al., 2006; Bengio & LeCun, 2007; Radford et al., 2019). In settings where explicit supervision is scarce or incomplete, embedding-based approaches provide a flexible abstraction for organizing, exploring, and querying large datasets. Recent foundation models trained on la rge astronomical or vision datasets produce rich and reusable embeddings that can be computed once and reused across multiple downstream tasks (Walmsley et al., 2022; Parker et al., 2025; Siudek et al., 2025). These representations offer a natural substrate for anomaly detection at survey scale, as rarity can be assessed directly in embedding space through density estimation or neighborhood structure. However, foundation models differ substantially in architecture, training objective, and modality handling, raising the question of how these design choices influence which objects are deemed anomalous (Ronecker et al., 2025).

This work addresses three complementary questions about anomaly detection with pretrained multimodal representations in large astronomical surveys. First, **to what extent do different foundation models agree on which objects are anomalous when applied to the same dataset?** More specifically, we ask whether anomaly rankings are largely model-agnostic or whether they depend strongly on architectural choices, training objectives, and modality handling. Second, **how does multimodality affect anomaly discovery?** We investigate whether combining imaging and spectroscopy highlights objects that would not be detected using unimodal representations alone, and whether cross-modal information helps separate astrophysical outliers from modality-specific artefacts. Third, **do embeddings that are effective for anomaly detection also encode known physical parameters in a structured and usable way?** This question connects unsupervised anomaly signals with downstream predictive behavior, providing an indirect probe of representation quality.

Our contributions are fourfold: (i) we present a scalable, fully unsupervised pipeline for multimodal anomaly detection in a joint Euclid-DESI imaging and spectroscopy dataset using pretrained foundation model embeddings; (ii) we perform a systematic cross-model comparison of anomaly rankings, highlighting both agreements and representation-specific differences; (iii) we combine density-based outlier detection with cross-modal misalignment signals to reduce contamination from unimodal artefacts and surface physically coherent candidates; and (iv) we provide qualitative case studies of high-ranking anomalies, illustrating both astrophysical systems of interest and recurrent instrumental or processing failure modes.

We emphasize that our goal is not to provide a definitive quantitative benchmark of anomaly discovery performance, since curated anomaly labels are not available for this Euclid–DESI subset.

Instead, we position this work as a representation-aware comparative study of how different foundation model embeddings structure rarity, cross-modal agreement, and anomaly prioritization in a realistic multimodal survey setting.

## 2 RELATED WORK

**Anomaly detection in astronomical surveys.** Unsupervised and weakly supervised anomaly detection has a long history in astronomy, initially relying on hand-crafted features combined with classical methods such as clustering, one-class SVMs, Isolation Forests, or self-organizing maps (Baron, 2019; Kurcz et al., 2016). With the advent of deep learning, generative approaches based on autoencoders and variational autoencoders have been widely explored, using reconstruction error or latent-space structure to identify rare objects (Margalef-Bentabol et al., 2020; Cheng et al., 2020; Storey-Fisher et al., 2020). While successful in recovering interacting galaxies or strong lens candidates, these methods often suffer from limited scalability, sensitivity to domain shift (e.g., PSF, depth, noise), and a tendency to absorb rare phenomena into the learned notion of normality. More recently, diffusion models and probabilistic generative transformers have been proposed for anomaly detection (Liu et al., 2025), though their application to large-scale astronomical surveys remains largely exploratory.

**Foundation models and embedding-based discovery.** An alternative paradigm relies on large self-supervised or weakly supervised foundation models that produce reusable embeddings capturing high-level morphological and physical information (Bommasani et al., 2022). In astronomy, models such as Zoobot (Walmsley et al., 2022; Willett et al., 2013), AstroPT (Siudek et al., 2025), and multimodal transformers such as AION (Parker et al., 2025) have demonstrated that learned representations can structure galaxy populations, support data-efficient downstream tasks, and facilitate unsupervised exploration of rare systems. Embedding-based anomaly detection has been shown to reveal visually distinctive but unlabeled subclasses through density estimation or nearest-neighbor analysis in latent space (Ronecker et al., 2025). However, how different foundation models disagree on what constitutes an anomaly, and whether multimodal embeddings systematically improve anomaly discovery relative to unimodal representations, remain open questions. This work addresses these gaps through a comparative, representation-aware analysis of anomaly rankings across models and modalities.

## 3 DATA AND PROBLEM SETUP

Our objective is to identify anomalous astronomical sources in large multimodal datasets without relying on explicit labels. We focus on anomaly definitions that can be formulated directly in representation space and applied at scale.

Our study uses data from the Euclid Quick Data Release 1 (Q1), focusing on a subset of sources for which both Euclid imaging and matched spectroscopic observations from the Dark Energy Spectroscopic Instrument (DESI) are available. This matched sample enables a controlled multimodal analysis combining high-resolution space-based imaging with ground-based spectroscopy. The dataset used in this work is the publicly available `astroPT_euclid_Q1_desi_dr1_dataset`, released on Hugging Face at `https://huggingface.co/datasets/msiudek/astroPT_euclid_Q1_desi_dr1_dataset`.

We use Euclid Visible Instrument (VIS) images together with Near-Infrared Spectrometer and Photometer (NISP) observations in the $Y$, $J$, and $H$ bands. For each object, square image cutouts are extracted around the target coordinates, yielding four-band image tensors (VIS, $Y, J, H$). Then, each source is associated with a DESI spectrum providing flux measurements as a function of wavelength (e.g 1). These spectra offer complementary physical information that is largely independent from image morphology. Finally, catalog-level physical quantities are associated with the Euclid Q1 sources, including spectroscopic redshifts measured by DESI and other derived parameters, are used exclusively for downstream analysis and evaluation. They are not used in the computation of anomaly scores.

The final Euclid–DESI matched sample contains 39,850 objects , all with valid spectroscopic redshift estimates. The sample is dominated by galaxies, with a smaller fraction of quasars.

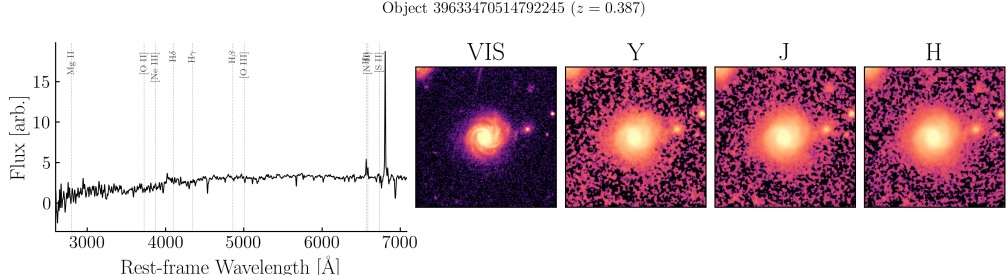

Figure 1: Example object from the Euclid–DESI matched dataset. Shown are the DESI spectrum (left) and Euclid imaging cutouts in the VIS, $Y$, $J$, and $H$ bands (right).

## 4 FOUNDATION MODELS AND EMBEDDINGS

We compare three foundation models that differ in architecture, training objective, and modality fusion strategy. Since our chosen models have not been originally pretrained with our dataset we apply lightweight adaptations to pre-trained models to make them compatible with Euclid Q1 and DESI data. This setting allows us to study how different representation learning paradigms shape anomaly detection behavior under realistic domain shift. The three models are not adapted in exactly the same way, because the minimum changes required to make them compatible with Euclid–DESI differ across architectures. As a result, our comparison should not be interpreted as a strictly controlled architecture-only ablation. Instead, it reflects a practical benchmark of reusable foundation-model pipelines under domain shift, where objective, architecture, and adaptation strategy jointly influence the resulting embeddings. Technical and implementation details are provided in the Appendices B.1, B.2 and B.3. Sources are available on github `https://github.com/MaxRonce/foundation-models-benchmark/`.

### 4.1 MODELS OVERVIEW AND ADAPTATION

**AstroPT** (Smith et al., 2024; Siudek et al., 2025) is a multimodal autoregressive foundation model based on the 'GPT' architecture (Radford et al., 2019) that is trained to model the joint distribution of astronomical observations across modalities. Its causal transformer architecture encourages embeddings to capture detailed statistical dependencies within and across images and spectra. In this work, we adapt AstroPT to the Euclid–DESI datset by retraining a single shared transformer on paired imaging and spectra, preserving the original autoregressive objective and generative inductive bias while enabling joint multimodal inputs. To ensure stable joint training despite strong modality imbalance, we introduce modality-wise loss reweighting so that image and spectral tokens contribute comparably to the training objective. This adaptation yields a unified representation space spanning imaging and spectroscopy. We preserve the original autoregressive objective and apply the minimum adaptation required for paired Euclid–DESI inputs, while noting that retraining on this specific domain may affect direct comparability with the other models.

**AstroCLIP** (Parker et al., 2024) is a cross-modal foundation model trained with a CLIP-style (Radford et al., 2021) contrastive objective, aligning image and spectrum embeddings in a shared latent space. Cosine similarity in this space directly reflects cross-modal agreement. We adapt AstroCLIP to Euclid Q1 by fine-tuning on matched Euclid–DESI pairs, primarily updating the image encoder, which was not originally trained on Euclid imaging, while preserving the original contrastive training objective. The spectrum encoder, pretrained on DESI data, is only lightly unfrozen during fine-tuning to stabilize cross-modal alignment and help clipping. We preserve the original contrastive objective and apply only the minimum fine-tuning needed to align Euclid images with DESI spectra, which may influence the resulting alignment geometry.

**AION** (Parker et al., 2025) is a multimodal foundation model based on the '4M' architecture (Mizrahi et al., 2023) that is designed to encode astronomical images and spectra into a unified representation optimized for downstream physical parameter prediction. It relies on modality-specific

tokenization pipelines followed by a shared transformer. A distinctive aspect of AION is its use of a discrete image tokenization based on a Vector-Quantized Variational Autoencoder (VQ-VAE). Since full retraining is computationally expensive, we use a lightweight U-Net adapter to transform Euclid images into representations that match the statistical properties of the image domain originally used to train the VQ-VAE. While this approach sacrifices some of the high spatial resolution information present in Euclid images, it enables controlled reuse of AION embeddings without retraining the full model. This lightweight reuse strategy avoids full retraining, but the adapter may alter or remove some spatial information from the original Euclid images.

## 4.2 Extracted Embeddings

For each foundation model, we extract one or more embedding representations aligned with the modalities supported by the architecture: (i) image-only embeddings, (ii) spectrum-only embeddings, and (iii) joint (multimodal) embeddings when available. To avoid arbitrary layer choices, we follow the extraction protocol recommended in the model's original paper or official implementation—typically using the final pooled representation or the last hidden state before the task-specific head, so as to obtain stable features that remain comparable across architectures.

Unimodal embeddings are obtained by forwarding a single modality through the corresponding encoder and applying the model's standard pooling strategy. In AstroPT, multimodal embeddings are produced by a joint autoregressive transformer operating on interleaved image and spectrum tokens, yielding a single representation that explicitly models cross-modal dependencies. In AION, modality-specific tokens are jointly processed by a shared transformer, resulting in a unified embedding per object. In AstroCLIP the multimodal representation of an object is defined by the paired image and spectrum embeddings associated with the same source. Cross-modal information is therefore captured through geometric proximity in the shared space.

Each embedding view is treated as an independent representation space and serves as the sole input to all anomaly detection and evaluation procedures.

## 5 Methods

### 5.1 Density-based outlier scoring and multimodal misalignment

We adopt a *density-based* anomaly criterion: objects whose embeddings lie in low-density regions of representation space are assigned higher anomaly scores. For each foundation model, we consider the available embedding modalities (image-only, spectrum-only, and joint when applicable) and train a separate density model for each (`model, modality`) pair.

Embedding densities are estimated using Masked Autoregressive Flows (MAFs) (Papamakarios et al., 2017), which provide exact and scalable log-likelihood evaluation in moderately high-dimensional spaces. After training, each object is scored by its negative log-likelihood under the learned density. These scores are standardized across the dataset to obtain a dimensionless anomaly measure, and objects are ranked accordingly, with higher scores indicating lower-density and more anomalous regions of the embedding space. To ensure comparability across models and modalities, all analyses rely on standardized rankings rather than raw likelihood values.

We isolate *multimodal* anomalies by combining unimodal rarity with explicit image–spectrum misalignment.

For each object, we compute percentile-based anomaly scores from image embeddings ($p_{\mathrm{img}}$), spectral embeddings ($p_{\mathrm{spec}}$), and a cross-modal mismatch score based on low cosine similarity ($p_{\mathrm{mis}}$). We prioritize candidates using a simple fusion score: $s_{\mathrm{mm}} = p_{\mathrm{mis}} \sqrt{p_{\mathrm{img}} \, p_{\mathrm{spec}}}$. This construction suppresses purely unimodal outliers while favoring anomalies supported by both modalities. In the appendix, we compare this geometric fusion to simpler alternatives, including arithmetic averaging, mismatch-only, and density-only rankings. We also compare our MAF-based anomaly scores to simpler baselines based on $k$-nearest-neighbor distance and Gaussian mixture models. These comparisons are intended as robustness checks rather than as exhaustive method selection.

## 5.2 CROSS-MODEL RANKING TRANSFER

To quantify this agreement and characterize systematic differences between representations, we made a *cross-model ranking transfer* analysis.

For a fixed embedding type (image-only, spectrum-only, or joint), each model assigns to every object an anomaly rank $r_i^{(m)} \in \{1, \ldots, N\}$, where lower ranks correspond to more anomalous samples. Ranks are derived independently for each (`model, embedding_key`) pair as described in Section 5.1. To ensure a fair comparison, we restrict the analysis to objects present in all models for the considered embedding type.

**Definition 5.1** (Ranking transfer). Given two models $A$ (source) and $B$ (target), we select the top $p\%$ most anomalous objects according to model $A$:

$$\mathcal{S}_A(p) = \{i \mid r_i^{(A)} \leq pN\}. \tag{1}$$

For each object $i \in \mathcal{S}_A(p)$, we then examine its rank under model $B$, $r_i^{(B)}$, and convert it to a percentile rank $100 \times r_i^{(B)}/N$. The resulting distribution measures how objects deemed anomalous by model $A$ are positioned in the ranking induced by model $B$.

**Definition 5.2** (Retention). To summarize the transfer behavior, we compute *retention rates* at fixed thresholds. In particular, we report the fraction of objects in $\mathcal{S}_A(1\%)$ that also fall within the top $1\%$ or top $10\%$ of model $B$'s ranking:

$$\text{Retain}_{A \to B}(q) = \frac{|\{i \in \mathcal{S}_A(1\%) \mid r_i^{(B)} \leq qN\}|}{|\mathcal{S}_A(1\%)|}, \qquad q \in \{1\%, 10\%\}. \tag{2}$$

High retention indicates strong agreement between models on what constitutes an extreme outlier, whereas low retention highlights representation-specific notions of anomaly.

We apply the ranking-transfer procedure between foundation models in both directions, treating each model in turn as a source of anomalies and evaluating how these objects are ranked by the others. In addition, we extend this analysis across modalities by comparing rankings obtained from image-only, spectrum-only, and joint embeddings. This joint cross-model and cross-modality evaluation avoids assuming a privileged reference model or modality and enables a representation-aware characterization of agreement and disagreement patterns. The procedure is invariant to monotonic rescalings of anomaly scores and does not require likelihoods to be comparable across models or modalities. Ranking transfer does not assume a shared density scale across models and is invariant to monotonic transformations of anomaly scores. It therefore provides a representation-aware diagnostic of cross-model agreement, revealing whether different foundation models reveal similar objects as outliers.

As we anomaly detection, we briefly assess how different embeddings encode known physical parameters using lightweight predictive probes. This *proxy evaluation* does not aim at measuring downstream task performance, but at characterizing representation geometry, including linear decodability, non-linearity, and how predictive information is distributed across embedding dimensions. Specifically, we predict selected physical quantities from image-only, spectrum-only, and joint embeddings using simple linear and non-linear models, and summarize differences across foundation models. Full definitions and metrics are provided in Appendix D.

## 6 RESULTS

### 6.1 LATENT STRUCTURE AND PHYSICAL GRADIENTS

Figure 2 visualizes 2D UMAP projections of the joint image–spectrum embeddings, colored by spectroscopic redshift. All three models recover a clear global redshift organization, showing that physically relevant structure is present in the latent space, but the geometry differs markedly across representations.

AstroPT and AION yield compact, relatively continuous manifolds with smooth redshift gradients. In contrast, AstroCLIP is more fragmented, with larger gaps and a broader spread, and shows a

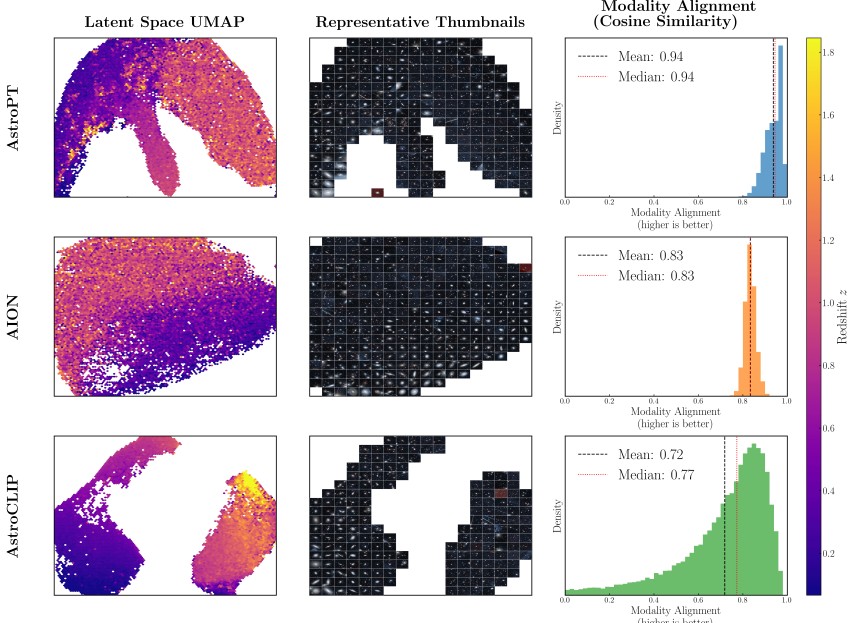

Figure 2: **Latent structure of multimodal embeddings.**
Left: UMAP projections of joint image–spectrum embeddings for AstroPT, AION, and AstroCLIP, colored by spectroscopic redshift. A smooth color gradient indicates that nearby points tend to share similar redshift. Middle: thumbnail mosaics sampled from the same projections, showing that local neighborhoods also correspond to coherent morphology. Right: distributions of image–spectrum cosine similarity for matched pairs; lower similarity indicates stronger cross-modal mismatch. Together, these panels illustrate that the three models encode physically meaningful structure, but with different global geometries and alignment profiles.

clearer separation between low- and high-redshift objects. This is consistent with its wider distribution of image–spectrum cosine similarities, including a tail toward low similarity: a contrastive objective enforces *relative* alignment (matched vs. unmatched) but does not necessarily constrain all matched pairs to have uniformly high absolute similarity, so weakly aligned pairs can remain farther apart in the embedding space.

Thumbnail mosaics over the projections (Figure 2) show that local neighborhoods correspond to coherent morphology, and that moving along the manifold tracks smooth changes in size, concentration, and color. This supports local embedding operations (density estimation, nearest-neighbor retrieval) as practical tools for large-scale exploration. A higher-resolution version of this figure is provided in the appendix (Figure 8).

## 6.2 CASE-BASED INSPECTION OF HIGH-RANKING ANOMALIES

In the absence of curated anomaly labels, we perform a *case-based inspection* of a small number of high-ranking candidates. Specifically, we examine representative objects selected from the top of the anomaly rankings, focusing on visually and spectroscopically unusual sources. This analysis is qualitative and illustrative, and does not aim at statistical validation.

Figure 3 shows a subset of such high-ranking anomalies. The examples are intentionally selected to highlight rare or peculiar systems that recur across different rankings, rather than to estimate class frequencies or detection performance.

**Observed anomaly patterns.** Visual inspection reveals a small number of recurring qualitative patterns. The categories below are descriptive summaries of commonly observed behaviors, rather than a strict or exhaustive classification.

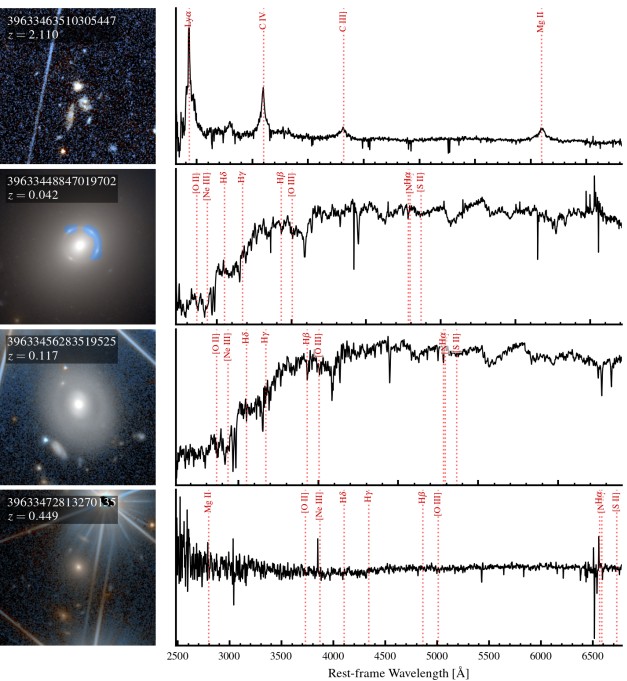

Figure 3: **Representative rare and peculiar galaxies identified as anomalies.**
Selected high-ranking candidates showing Euclid image cutouts and DESI spectra. These examples
illustrate the diversity of rare astrophysical systems and data-quality outliers surfaced by the
anomaly detection pipeline. Each row shows one Euclid image cutout and its matched DESI
spectrum for an object selected from the top-ranked candidates of our embedding-based anomaly
scores.

*AGN-like spectral outliers.* Several candidates exhibit broad or asymmetric emission lines that deviate strongly from the bulk of the DESI sample. These rare spectral signatures are primarily highlighted by spectrum-based anomaly scores.

*Strong gravitational lens candidates.* Some high-ranking anomalies display arc-like or ring-like morphologies in Euclid imaging, combined with spectra that are difficult to interpret as a single galaxy. These systems are typically emphasized by multimodal embeddings, suggesting a genuinely cross-modal anomaly signal.

*Post-starburst and quiescent galaxies.* A subset of image-driven anomalies corresponds to smooth, early-type morphologies whose spectra show strong Balmer absorption and weak or absent emission lines, consistent with rare or transitional evolutionary states.

*Instrumental or processing artefacts.* Diffraction spikes, saturation effects, ghosts, and residual image defects are frequently detected as strong image-only outliers. While not of direct astrophysical interest, these cases serve as a useful control, illustrating how multimodal information can downweight purely instrumental anomalies.

### 6.3 CROSS-MODEL RANKING TRANSFER

We summarize cross-model agreement using *retention rates* at 1% and 10% in Table 1 (see Section 5.2 for definitions). The full percentile-rank histograms, including both transfer directions overlaid for each model pair, are provided in Appendix C (Figure 5).

Across all modalities, we find that a non-negligible fraction of the most extreme outliers (top 1%) is shared between models, while the broader top-10% agreement can be strongly direction-dependent, reflecting representation-specific re-ordering beyond the very top ranks. In particular, AION and AstroPT show the strongest overall agreement, especially for spectra where AION→AstroPT retains

Table 1: **Cross-model ranking transfer (retention rates).** Fraction of top anomalies selected by a source model (top 1%) that remain in the top 1% and top 10% under a target model, reported separately for image-only, spectrum-only, and joint embeddings. Format: $A \rightarrow B$ % ($A \leftarrow B$ %). For example, a top-1 retention of 21.6 means that 21.6% of objects ranked in the most anomalous 1% by one model are also ranked in the most anomalous 1% by the other. Retention at 1% reflects agreement on the most extreme anomalies, while retention at 10% captures broader overlap despite ranking differences. High top-10 but lower top-1 retention suggests that two models detect similar unusual objects but prioritize them differently.

| Embedding | Pair | Top 1% retained | Top 10% retained |
|---|---|---|---|
| Images | AION $\leftrightarrow$ AstroPT | 21.6 (21.6) | 47.2 (64.3) |
| Images | AstroPT $\leftrightarrow$ AstroCLIP | 16.8 (16.8) | 43.0 (58.0) |
| Images | AstroCLIP $\leftrightarrow$ AION | 7.3 (7.3) | 32.9 (23.4) |
| Spectra | AION $\leftrightarrow$ AstroPT | 9.3 (9.3) | 79.1 (55.8) |
| Spectra | AstroPT $\leftrightarrow$ AstroCLIP | 14.6 (14.6) | 41.5 (44.5) |
| Spectra | AstroCLIP $\leftrightarrow$ AION | 3.8 (3.8) | 33.9 (10.1) |
| Joint | AION $\leftrightarrow$ AstroPT | 19.1 (19.1) | 58.5 (31.7) |
| Joint | AstroPT $\leftrightarrow$ AstroCLIP | 11.6 (11.6) | 35.7 (38.9) |
| Joint | AstroCLIP $\leftrightarrow$ AION | 5.3 (5.3) | 23.1 (25.4) |

79.1% of AION top-1% objects within AstroPT's top-10%. Conversely, agreement between AION and AstroCLIP is consistently lower at the extreme tail (top-1%), and exhibits pronounced asymmetries at top-10% in several modalities (notably Spectra). Joint embeddings often show retention rates and percentile-rank distributions that lie between the image-only and spectrum-only settings, indicating a mixed agreement regime where cross-model overlap increases relative to images but remains below spectra in term of transfers. Overall, these results indicate that while the *ordering* of anomalies varies across foundation models, a meaningful subset of the most extreme outliers remains shared, and directional retention at 10% helps reveal systematic differences in anomaly prioritization.

## 6.4 PROXY EVALUATION OF EMBEDDING STRUCTURE

As a complementary diagnostic, we evaluate how different embeddings encode known physical parameters using lightweight predictive probes (Appendix D). While not a measure of anomaly detection performance, this analysis helps interpret differences in embedding geometry across models.

We stress that these predictive probes are not intended as direct measures of anomaly-detection performance. Rather, they serve as complementary diagnostics of embedding geometry, indicating how predictive information is organized, how linearly decodable it is, and how this organization differs across models and modalities.

Overall, spectrum-based embeddings achieve the strongest predictive accuracy, reflecting both the high information content of DESI spectra and their closer match to pretrained encoders. Joint embeddings do not uniformly outperform unimodal ones, but instead redistribute predictive signal across dimensions, depending on the model architecture. Finally, on our proxy tasks we find a larger linear-to-nonlinear probe gap for AstroPT than for AstroCLIP 2, meaning that a non-linear probe extracts more additional predictive signal from AstroPT embeddings. We do not interpret this as a general property of autoregressive models, in fact autoregressive transformers can exhibit strongly linear internal representations as mentionned in Nanda (2023), but as a model, and extraction-specific effect in our setting (architecture, pooling, and adaptation).

This interpretation is consistent with prior work showing that contrastive objectives shape embedding geometry in ways that can make simple readouts effective: analyses of contrastive losses link them to alignment of positives and uniformity of representations, which together promote well-structured feature spaces (Wang & Isola, 2022; Wang & Liu, 2021). Conversely, likelihood-trained transformer embeddings have been reported to exhibit strong anisotropy (Ethayarajh, 2019), and follow-up work suggests that related representation-geometry issues can also be driven by systematic biases rather than anisotropy alone (Fuster Baggetto & Fresno, 2022).

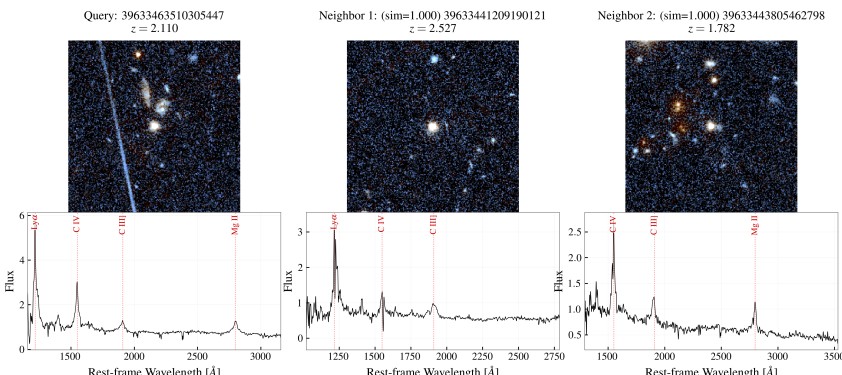

Figure 4: **Similarity search around an AGN anomaly.** One-row visualization showing a query object (left) and its nearest neighbors retrieved in one embedding spaces, all neighbors shows consistent AGN-like spectral and morphological properties.

## 6.5 SIMILARITY SEARCH AROUND DISCOVERED ANOMALIES

Beyond isolated detection, embedding spaces enable a natural extension of anomaly discovery through *similarity search*. Starting from a single high-confidence anomaly, nearest-neighbor queries allow the construction of homogeneous candidate sets that share related representation-level features. This behavior highlights a two-stage discovery workflow enabled by foundation model embeddings: rare objects are first identified through density-based outlier detection, and then expanded into physically meaningful candidate sets via similarity search. Importantly, this process remains fully unsupervised and scales naturally to larger survey volumes.

## 7 DISCUSSION AND OUTLOOK

This study highlights a central aspect of embedding-based anomaly detection: anomalies are inherently *representation-relative*. When applied to the same Euclid–DESI sample, different foundation models identify largely overlapping sets of extreme outliers, but assign them different priorities depending on architecture, training objective, and modality handling. These differences are not arbitrary; they reflect distinct inductive biases and representation geometries.

Multimodality does not uniformly improve anomaly detection, but it changes its failure modes. Unimodal density tails, particularly in image space, are often dominated by instrumental artefacts, whereas candidates supported by multiple signals, moderate rarity in both modalities or explicit cross-modal misalignment, are more likely to remain coherent under joint inspection. In this sense, multimodal information acts primarily as a filter and prioritization mechanism rather than a single best score.

Beyond isolated detection, embeddings naturally support similarity search, enabling the expansion of a small number of high-confidence anomalies into homogeneous candidate sets without supervision. This two-stage workflow outlier detection followed by neighborhood exploration is lightweight, scalable, and well suited to large survey data.

Several limitations remain. Our analysis is restricted to a Euclid–DESI matched subset and relies on proxy evaluation in the absence of ground-truth anomaly labels. Domain adaptation is another key bottleneck: Euclid image encoders were only lightly fine-tuned, and the AION case illustrates the difficulty of integrating new imaging domains into discrete tokenization pipelines without retraining large components.

Looking forward, scaling this approach to larger Euclid releases, performing controlled source injection tests, and improving multimodal calibration and image-domain adaptation will be essential for making embedding-based anomaly discovery robust at survey scale.

AUTHOR CONTRIBUTIONS

Marc Huertas-Company was the principal investigator (PI) and contributed to the scientific direction of the project. Alexandre Chanson was the university supervisor and provided technical supervision, with additional co-supervision from Malgorzata Siudek. Malgorzata Siudek also contributed to model-related work, including AstroPT experiments. Michael J. Smith contributed to the AstroPT-related development and experiments. Anna Preto led the AstroCLIP fine-tuning and the associated development efforts, with contributions from Clara Bonini. Julien Zoubian contributed to project organization and initiation, and supported the software infrastructure and Git-based workflow. Maxime Ronceray led the project execution, including embedding extraction, adaptation of AION, development of the anomaly detection methods, integration of all models into a unified pipeline, and manuscript writing, with feedback from all authors.

ACKNOWLEDGMENTS

This work was carried out in the context of AstroInfo 2025, which includes a CNRS thematic school and a hackathon supported by AISSAI. We gratefully acknowledge their respective contributions, as well as the AstroInfo organizing committee for the coordination and support that made these activities possible. We gratefully acknowledge support from the CNRS/IN2P3 Computing Center (Lyon - France) for providing computing and data-processing resources needed for this work. We also thank François Lanusse and Liam Parker for their technical support and helpful discussions regarding the AION model.

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

## APPENDIX

## A PREPROCESSING DETAILS

This appendix describes the preprocessing steps applied to Euclid imaging and DESI spectroscopy prior to embedding extraction. These steps are intentionally described briefly in the main text, as they do not constitute methodological contributions of this work.

### A.1 IMAGE PREPROCESSING

Euclid VIS and NISP images are first converted to floating-point representation. Invalid pixel values (NaNs or infinities) are replaced with zeros. Pixel intensities are converted from analog-to-digital units (ADU) to flux density units following Euclid photometric calibration conventions:

$$f_\nu = \text{ADU} \times \frac{\text{ZP}_\nu}{3631},$$

where $\text{ZP}_\nu$ denotes the band-specific zero point.

The four bands (VIS, $Y$, $J$, $H$) are stacked channel-wise to form a four-channel image tensor. Image cutouts are resized to a fixed spatial resolution using bilinear interpolation to match the input requirements of the different foundation models. No data augmentation is applied, in order to preserve astrophysical and instrumental structures relevant for anomaly detection.

### A.2 SPECTRAL PREPROCESSING

DESI spectra are converted to continuous flux representations and, when required, interpolated onto a common wavelength grid. Each spectrum is normalized to reduce scale variations across objects while preserving relative spectral shape. This normalization mitigates sensitivity to absolute flux levels and facilitates stable embedding extraction across the sample.

## B ADDITIONAL IMPLEMENTATION DETAILS

This appendix provides technical details that are omitted from the main paper for clarity and space constraints. These details are included to ensure reproducibility and to document the specific adaptations applied to each foundation model.

### B.1 ASTROPT: EUCLID–DESI ADAPTATION

We follow the AstroPT framework of Smith et al. (2024), which was subsequently adapted to Euclid imaging by Siudek et al. (2025), and extend it here to a multimodal setting by jointly modeling paired Euclid images and DESI spectra within a single autoregressive transformer.

**Inputs.** Euclid images are formatted as three-channel composites resized to $224 \times 224$ with bilinear interpolation. Pixel NaNs and infinities are sanitized and values scaled to [0,1] prior to patchification. DESI spectra are padded or truncated to a fixed length of 7781 samples and normalized by their per-spectrum standard deviation to reduce amplitude variation while preserving spectral shape.

**Tokenization and architecture.** Images are tokenized into non-overlapping 16×16 patches, yielding 196 tokens per image, while spectra are segmented into fixed-length patches of size 10, producing approximately 779 tokens per spectrum. Tokens from both modalities are processed by a shared causal transformer with 12 layers, 12 attention heads, and embedding dimension 768, using a maximum context length of 1024 tokens.

**Training objective and modality balancing.** Images are trained using a standard autoregressive next-token prediction objective, while spectra are trained using full-sequence reconstruction. To prevent the spectral modality from dominating optimization due to its larger token count, per-modality loss terms are reweighted inversely proportional to the number of tokens in each modality, ensuring comparable aggregate contributions from images and spectra during training.

**Training details.** Optimization is performed using AdamW with learning rate 6×10-4, weight decay 0.1, and gradient clipping at 1.0. Training uses mixed-precision arithmetic (bfloat16) and an effective batch size of 32 achieved via gradient accumulation. A linear learning-rate warmup is followed by cosine decay, and training is run until convergence on the paired Euclid-DESI dataset.

### B.2 ASTROCLIP: EUCLID FINE-TUNING

AstroCLIP is adapted to Euclid Q1 imaging by fine-tuning a public AstroCLIP checkpoint on matched Euclid–DESI pairs using the original contrastive objective.

**Fine-tuning strategy.** The image encoder is fine-tuned on Euclid cutouts resized to $144 \times 144$ to account for domain shift in imaging data. Unless stated otherwise, the spectrum encoder is also allowed to update during fine-tuning to maintain alignment between modalities.

**Spectral preprocessing.** DESI spectra are padded or truncated to a fixed length of 7700 samples and normalized using z-score standardization prior to encoding.

**Optimization.** Fine-tuning is performed for 5 epochs using AdamW with learning rate $3 \times 10^{-6}$ and weight decay $5 \times 10^{-4}$. Mixed-precision training and gradient accumulation are used, yielding an effective batch size of $128 \times 2$. The contrastive temperature parameter is learnable during training.

### B.3 AION: EUCLID ADAPTATION AND VQ-VAE COMPATIBILITY

AION relies on a discrete image tokenization scheme based on a Vector-Quantized Variational Autoencoder (VQ-VAE), in which images are mapped to sequences of discrete codebook indices prior to multimodal transformer processing. This design introduces specific challenges when adapting the model to a new imaging survey such as Euclid.

**Limitations of direct modality insertion.** A naive adaptation strategy consists in introducing Euclid imaging as a new modality by duplicating existing projection layers and fine-tuning only these layers while keeping the VQ-VAE encoder and transformer frozen. This approach implicitly assumes that Euclid images are linearly projectable into the latent space learned for ground-based surveys.

In practice, this assumption does not hold. Differences in point-spread function, noise statistics, and spatial resolution lead to discrete token distributions that are poorly aligned with the codebook regions expected by the pretrained transformer, resulting in degraded representations.

**Lightweight adapter strategy.** Fully fine-tuning the VQ-VAE encoder jointly with the transformer would resolve this mismatch but is computationally prohibitive in the lightweight adaptation regime considered here.

Instead, we introduce a lightweight U-Net adapter trained to transform Euclid images into representations that match the statistical properties of the image domain originally used to train the VQ-VAE. The adapted images are then passed through the frozen VQ-VAE encoder, producing discrete tokens compatible with the pretrained transformer.

This strategy enables the use of Euclid data within AION without retraining the full model. While it sacrifices some Euclid-specific imaging advantages (notably resolution), it preserves compatibility with the learned discrete representation and allows controlled analysis of downstream embeddings.

## C    CROSS-MODEL RANKING TRANSFER DISTRIBUTIONS

Figure 5 reports the full percentile-rank distributions associated with the cross-model ranking transfer experiment summarized in Table 1. For each embedding view (rows: Images, Spectra, Joint) and each model pair (columns), we overlay the two transfer directions in the same panel: $A \to B$ and $B \to A$. Each histogram shows the percentile ranks, under the target model, of the objects selected in the top 1% of the source model.

Vertical dashed lines mark the target top 1% and top 10% thresholds (1 and 10 in percentile units), and the legend reports the corresponding retention rates in the format $A \to B$ % ($A \leftarrow B$ %), i.e. the fraction of the source top-1% that remains within the target top-1% or top-10%.

These distributions provide a diagnostic view of how anomalies are re-ordered across models. A strong concentration toward low target percentiles indicates shared extreme outliers, while long tails reveal representation-dependent prioritization.

### C.1    EXTENDED SIMILARITY SEARCH EXAMPLES

To further illustrate the behavior of similarity search across models and modalities, we provide an extended visualization of nearest-neighbor retrievals for the same AGN candidate discussed in Section 6.5.

Figure 6 shows similarity search results obtained independently from AION, AstroPT, and Astro-CLIP embeddings, and for each embedding type (image-only, spectrum-only, and joint). While the overall behavior is consistent with the summary shown in the main text, the extended visualization reveals model-specific nuances.

Spectrum-based embeddings across all models prioritize similarity in emission-line structure and continuum shape, leading to tight redshift and spectral-feature coherence. Image-based embeddings retrieve objects with comparable compactness and surface-brightness profiles, but may include morphologically similar non-AGN contaminants. Joint embeddings balance both effects, retrieving candidates that are simultaneously consistent in morphology and spectral features.

These examples further support the interpretation that similarity search is best understood as a representation-conditioned operation: retrieved neighbors reflect the inductive biases and modality weighting of the underlying embedding. Nevertheless, for physically distinctive systems such as AGN, this variability remains bounded, and all models recover scientifically meaningful candidate sets.

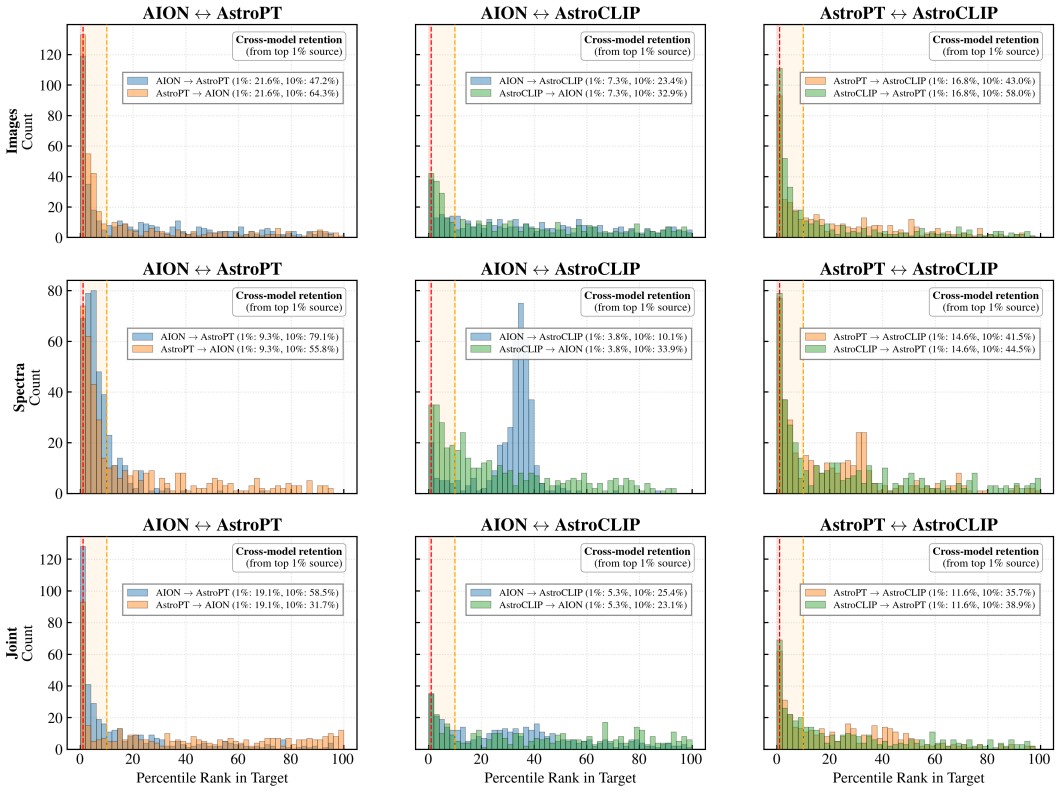

Figure 5: **Bidirectional percentile-rank distributions for cross-model ranking transfer.** Rows correspond to embedding views (Images, Spectra, Joint). Columns correspond to unordered model pairs, with both transfer directions overlaid. For each direction, we select the source model's top 1% anomalies and plot their percentile ranks under the target model.

# D PROXY EVALUATION: PREDICTIVE STRUCTURE OF EMBEDDINGS

## D.1 MOTIVATION

In the absence of dense ground-truth labels, we use lightweight predictive probes as a proxy to characterize the structure of learned embeddings. The goal is not to optimize downstream performance, but to assess how physical information is encoded: linearly or non-linearly, compactly or in a distributed manner.

## D.2 PREDICTION TASKS AND MODELS

We predict available physical parameters (e.g., spectroscopic redshift and derived quantities) from image-only, spectrum-only, and joint embeddings. Two models are used: a linear Ridge regressor probing linear decodability, and a non-linear gradient-boosted decision tree (LightGBM), capturing more complex dependencies. Performance is measured using $R^2$ and MAE.

## D.3 LINEAR DECODABILITY GAP

We define the *linear decodability gap* as

$$\Delta_{\mathrm{lin}} = R^2_{\mathrm{LGBM}} - R^2_{\mathrm{Ridge}},$$

which quantifies predictive information that is only accessible through non-linear transformations.

Table 2: **Proxy evaluation summary** (mean±std across predicted parameters). We report $R^2$ (LightGBM), Predictive Ratio (PR), Predictive Efficiency Score (PES), Coverage-Weighted Performance (CWP), and the linear decodability gap $\Delta_{\text{lin}} = R^2_{\text{LGBM}} - R^2_{\text{Ridge}}$.

| Model | Embedding | $R^2$ | PR | PES | CWP | $\Delta_{\text{lin}}$ |
|---|---|---|---|---|---|---|
| AION | image | 0.10±0.15 | 0.15±0.08 | 1.47±2.12 | 0.008±0.018 | 0.09±0.29 |
| AION | joint | 0.10±0.20 | 0.15±0.09 | 1.60±2.21 | 0.003±0.034 | -0.00±0.05 |
| AION | spectra | 0.40±0.33 | 0.11±0.06 | 6.29±7.16 | 0.030±0.022 | 0.04±0.12 |
| AstroPT | image | 0.27±0.30 | 0.10±0.05 | 4.89±6.44 | 0.019±0.026 | 0.20±0.68 |
| AstroPT | joint | 0.41±0.29 | 0.05±0.03 | 13.04±14.68 | 0.016±0.009 | 0.15±0.54 |
| AstroPT | spectra | 0.38±0.30 | 0.08±0.05 | 8.34±9.77 | 0.022±0.012 | 0.14±0.53 |
| AstroCLIP | image | 0.40±0.24 | 0.08±0.04 | 7.63±7.20 | 0.024±0.016 | 0.03±0.09 |
| AstroCLIP | joint | 0.49±0.25 | 0.09±0.05 | 7.77±6.21 | 0.038±0.017 | 0.02±0.04 |
| AstroCLIP | spectra | 0.44±0.29 | 0.08±0.04 | 10.69±12.64 | 0.024±0.011 | 0.02±0.02 |

### D.4 EFFECTIVE PREDICTIVE DIMENSIONALITY

To quantify how predictive information is distributed across embedding dimensions, we compute dimension-wise feature importances using SHAP values. Aggregating these yields a global importance profile, from which we derive the *Predictive Ratio* (PR), a normalized participation ratio measuring concentration versus distribution of predictive signal. The *Effective Predictive Dimensionality* (EPD) is defined as $\text{EPD} = \text{PR} \times D$.

### D.5 DERIVED EFFICIENCY METRICS

To jointly capture accuracy and dimensional usage, we report two derived quantities:

$$\text{PES} = \frac{R^2}{\text{PR} + \varepsilon}, \qquad \text{CWP} = R^2 \times \text{PR},$$

which respectively favor compact and distributed predictive encodings.

### D.6 RESULTS

Table 2 reports aggregated results across predicted parameters. These metrics highlight systematic differences in representation geometry across models and modalities, complementing the anomaly-based analyses in the main text.

## E ROBUSTNESS CHECKS: ANOMALY-SCORING BASELINES AND FUSION ABLATIONS

To address the dependence of our conclusions on specific scoring choices, we perform two complementary robustness checks. First, we compare the anomaly rankings obtained with Masked Autoregressive Flows (MAFs) to simpler baseline scoring methods applied to the same embeddings. Second, we ablate the multimodal fusion rule used to combine image rarity, spectral rarity, and image–spectrum mismatch.

### E.1 BASELINE ANOMALY-SCORING METHODS

In the main paper, anomaly scores are derived from density estimation with MAFs. To test whether our ranking-based conclusions are tied to this specific estimator, we compare MAF rankings to two simpler alternatives computed on the same standardized embeddings.

For each embedding space, we compute the average cosine distance to the $k = 10$ nearest neighbors. We also fit a Gaussian mixture model (GMM) and score each sample by its negative log-likelihood under the fitted mixture.

For numerical stability, when the embedding dimensionality is large we first apply PCA before fitting the mixture model. In our implementation, we use up to 10 mixture components and reduce the dimensionality to at most 32 PCA components when needed.

## E.2 COMPARISON METRIC

Because the absolute scales of MAF, kNN, and GMM scores are not directly comparable, we compare methods using Spearman rank correlation between their anomaly rankings. This evaluation focuses on whether the different scoring rules prioritize similar objects as anomalous, independently of score calibration.

## E.3 FUSION ABLATION

In the main text, multimodal anomalies are prioritized using the fusion score

$$s_{\mathrm{mm}} = p_{\mathrm{mis}} \sqrt{p_{\mathrm{img}} \, p_{\mathrm{spec}}},$$

where $p_{\mathrm{img}}$, $p_{\mathrm{spec}}$, and $p_{\mathrm{mis}}$ are percentile-based anomaly scores derived from image rarity, spectral rarity, and cross-modal mismatch, respectively.

We compare this geometric aggregation to three simpler alternatives:

$$s_{\mathrm{arith}} = \frac{p_{\mathrm{img}} + p_{\mathrm{spec}} + p_{\mathrm{mis}}}{3}, \tag{3}$$

$$s_{\mathrm{mis}} = p_{\mathrm{mis}}, \tag{4}$$

$$s_{\mathrm{dens}} = \sqrt{p_{\mathrm{img}} \, p_{\mathrm{spec}}}. \tag{5}$$

The purpose of this ablation is to assess whether our conclusions depend strongly on the exact aggregation formula.

As in the baseline-method comparison above, we quantify similarity between rankings using Spearman rank correlation, here between the geometric fusion ranking and each alternative:

- geometric vs. arithmetic,
- geometric vs. mismatch-only,
- geometric vs. density-only.

## E.4 RESULTS

Figure 7 summarizes these robustness checks.

**MAF vs. simpler anomaly baselines.**   Across most foundation-model and embedding pairs, MAF and GMM rankings exhibit moderate-to-strong agreement. This is expected to some extent, since both methods define anomaly through low estimated density, although MAF provides a more flexible density model than a finite Gaussian mixture. We take this agreement as evidence that the main ranking patterns are not solely tied to the particular MAF parameterization. By contrast, agreement between MAF and cosine $k$NN is often weak and in several cases close to zero, indicating that local neighborhood-based scoring captures a substantially different notion of outlierness than global density estimation. Overall, these results suggest that our conclusions are stable across alternative density-based scorers, but not across fundamentally different anomaly criteria.

**Fusion ablation.**   The geometric fusion used in the main paper is highly correlated with a simple arithmetic average across all three foundation models, indicating that the resulting multimodal ranking is not highly sensitive to the precise form of the full fusion rule. By contrast, rankings based on mismatch only or density only deviate more substantially from the full multimodal score, especially for some models. We interpret this ablation as evidence that the key design choice is to combine rarity and cross-modal misalignment, rather than to rely on either component alone; within that combined setting, both arithmetic and geometric fusion lead to broadly similar rankings.

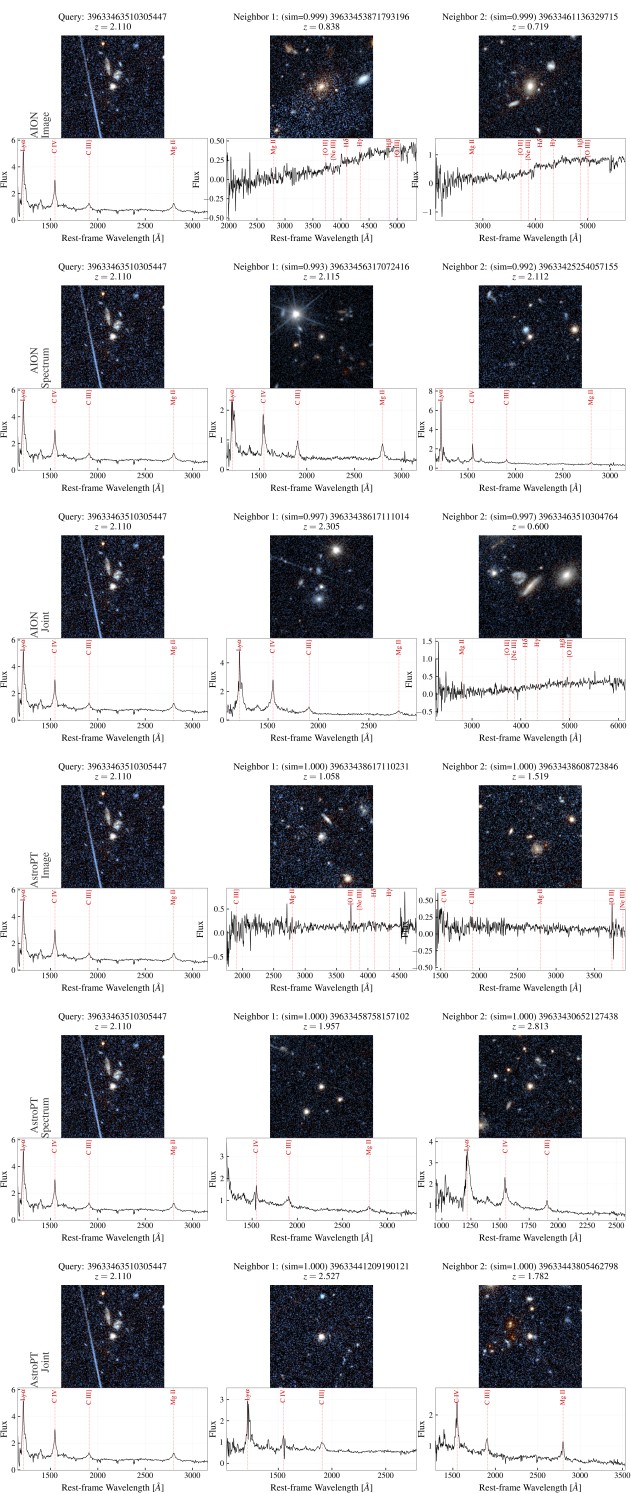

Figure 6: **Extended similarity search across models and modalities.** Nearest neighbors retrieved for the same AGN query using AION, AstroPT, and AstroCLIP embeddings, across image-only, spectrum-only, and joint representations. This expanded view highlights both consistent recovery of AGN-like systems and representation-specific retrieval biases.

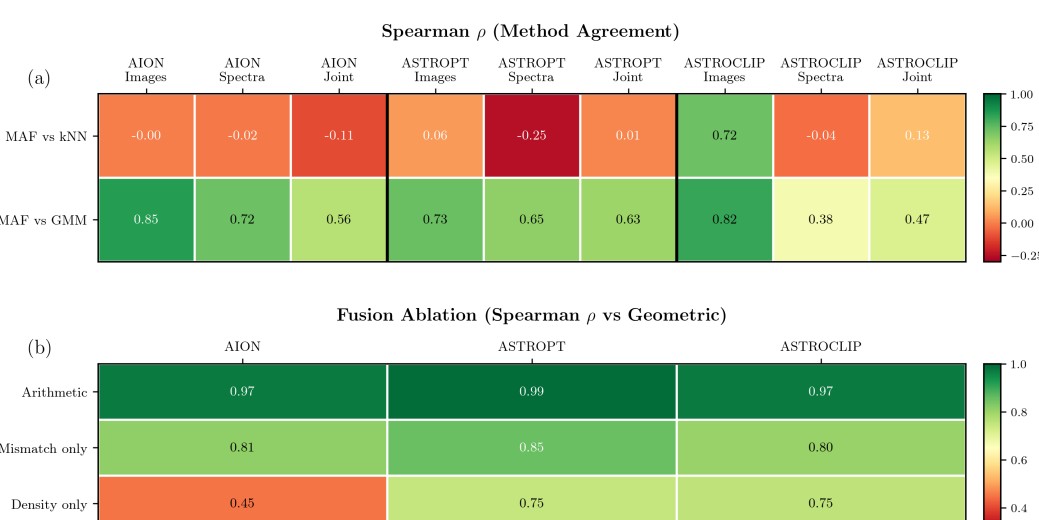

Figure 7: **Robustness checks for anomaly scoring and multimodal fusion. (a)** Spearman rank correlation between MAF-based anomaly rankings and two simpler baselines, cosine $k$NN distance and Gaussian mixture models (GMM), computed independently for each foundation model and embedding type. **(b)** Fusion ablation: Spearman rank correlation between the geometric multimodal fusion used in the main paper and three alternatives: arithmetic averaging, mismatch-only, and density-only. High correlation indicates that the corresponding ranking is similar to the one induced by the geometric score.

# AstroPT

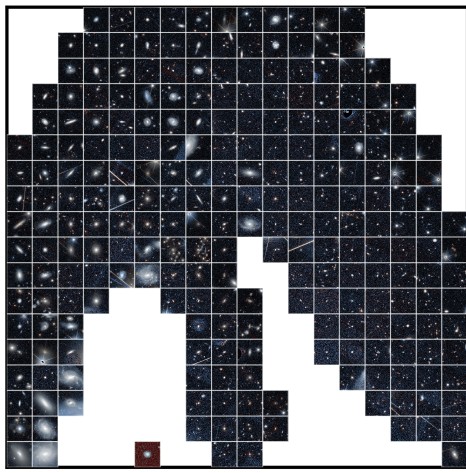

# AION

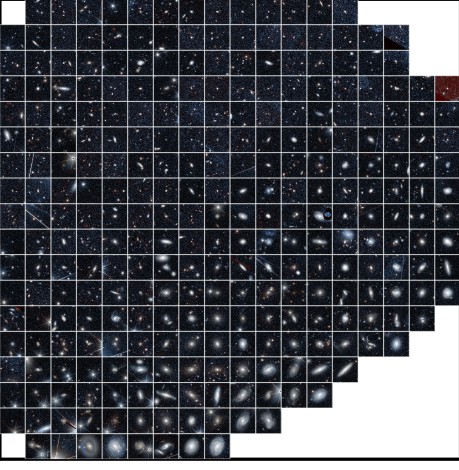

# AstroCLIP

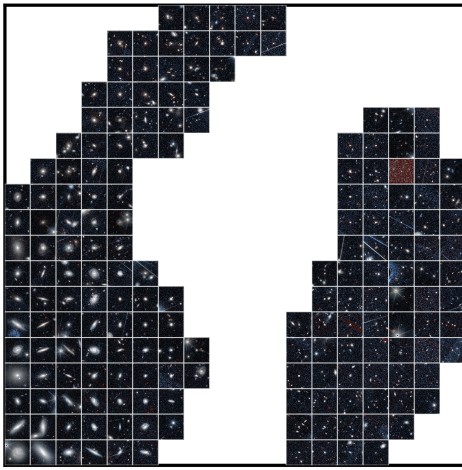

Figure 8: **Thumbnail mosaics sampled across the projections, illustrating coherent morphology and smooth physical gradients in local neighborhood** High-resolution version of Figure 2's 2nd collumn.

