# OpenReview forum: "Benchmarking foundation models for unsupervised discovery in large multimodal astrophysical datasets"
_ICLR.cc/2026/Workshop/FM4Science — ICLR 2026 Workshop FM4Science Poster_

### Official Review · Reviewer_FQJS · 2026-02-23
**Interesting Multimodal Analysis, but Limited as a General Benchmark**

**Rating:** 5
**Confidence:** 4

**Review:**

Summary

This paper studies how different multimodal foundation model embeddings behave for unsupervised anomaly discovery in astronomical survey data. Using a matched Euclid imaging and DESI spectroscopy dataset, the authors compare three representation paradigms (AstroPT, AstroCLIP, and AION) and introduce a pipeline combining density-based anomaly scoring with a cross-modal misalignment signal. A ranking-transfer analysis is used to compare anomaly lists across models, and qualitative inspection of high-ranking objects is presented to argue that multimodal signals help suppress instrumental artefacts.

The topic is relevant to the workshop and the choice of models is interesting. However, I found the empirical analysis less convincing than the paper’s framing suggests, and I am somewhat unsure how broadly the conclusions can be interpreted.

Strengths

1. Thoughtful selection of foundation models. The comparison spans autoregressive, contrastive, and multimodal predictive representations, which makes the study conceptually interesting rather than simply comparing model scale.

2. Strong domain motivation. The multimodal anomaly detection setup is well aligned with real scientific workflows, and the representation-relative perspective is relevant for deploying foundation models in practice.

3. Useful diagnostic idea. The ranking-transfer analysis provides an intuitive way to compare anomaly rankings across models without assuming calibrated likelihoods.

4. Clear presentation. The paper is generally easy to follow, and the figures help build intuition about the behavior of embeddings and anomaly rankings.

Major Concerns

1. Empirical analysis feels more exploratory than evaluative. Many conclusions rely on qualitative inspection of selected anomalies, and it is not always clear how strongly the claims follow from the experiments. For example, the argument that multimodal fusion reduces artefacts would benefit from quantitative validation rather than visual examples alone.

2. Limited evaluation scope. The study focuses on a single dataset and a single anomaly detection workflow, which makes it difficult to assess how general the findings are beyond this specific setting.

3. Selected examples are not clearly linked to specific methods or rankings. Section 6.2 discusses several categories of anomalies (e.g., AGN-like systems, lens candidates, artefacts), but it remains unclear which model, modality, or anomaly score produced each example. As a result, it is difficult to evaluate claims about representation-specific behavior or multimodal improvements, since the qualitative analysis reads more as an illustrative overview than a controlled comparison.

4. Uncontrolled comparison across models. The adaptation strategies differ substantially (retraining vs fine-tuning vs adapter-based alignment), making it hard to disentangle architectural effects from engineering choices.

5. Heuristic fusion score. The multimodal anomaly score is intuitive but introduced without ablation or sensitivity analysis, and alternative formulations are not explored.

6. Dependence on a single anomaly scoring method. The analysis relies on density estimation with flows, but it is unclear whether the main observations would hold under simpler or alternative anomaly metrics.

---

### Official Review · Reviewer_wxow · 2026-02-23
**A solid paper proposing a pipeline for unsupervised task multimodal evaluation among different foundation models**

**Rating:** 9
**Confidence:** 3

**Review:**

1. Summary

This solid paper with quite enough workload aims at the unsupervised task which is especially important for astronomical studies: anomaly detection. Due to the lack of "ground truth", this kind of tasks cannot have metrics for single model to be evaluated, like accuracy or MSE/MAE loss; so this paper focuses on the comparison between different models. The authors have taken advantage of the fact that foundation models rely on the concept of embeddings, to propose the metric of retention rates to represent cross-model ranking transfer, based on the density in latent space. This has perfectly avoided time-consuming human evaluation for anomalies, and found the result of AstroCLIP's mismatching with the other two astronomical foundation models. Overall, this is a good paper with both theoretical construction and example testing.

2. Strengths

- This paper has also shown the latent structure of multimodal embeddings for three models (figure 2), as a complement for the cross-model transfer ranking (table 1). They did the UMAP projections for the three models to a 2D plane, and consider two known quantities as "color bars": the redshift (left column) and the morphology (middle column). They found that AION and AstroPT's latent space is smoother while AstroCLIP's is more fragmental, while both features (no matter the redshift or the morphology) are not used during the Masked Autoregressive Flows training so "smoother" means better. This is a great way to validate their conclusion drawn by the cross-matching creterion.

- Besides the qualitative analysis for the latent space above, they also did a quantitative analysis, by using regression for redshift as a probe for the embeddings. In the appendix, they also mentioned the problem for unsupervised learning: there is no "ground-true labels" for the datasets, so they used the regression task to test whether the embeddings has learnt information. That is a more intuitive way than the UMAP projection way above.

3. Weaknesses & Critiques

- The predictive probe tasks (e.g., redshift regression) appear somewhat limited in their informativeness. Such tasks are already known to be well captured by modern foundation model embeddings, and the paper does not clearly demonstrate what additional insight is gained in this specific setting. Moreover, it is unclear how these proxy metrics relate to anomaly detection performance. Predictive accuracy reflects average structure in the data, whereas anomaly detection focuses on rare and atypical samples. As a result, the connection between the probe results and the main objective of the paper remains indirect. Finally, the lack of simple baselines (e.g., regression directly on raw inputs) makes it difficult to assess whether the embeddings provide a meaningful advantage in this context.

---

### Official Review · Reviewer_VGdg · 2026-02-24
**[Borderline Reject] An approach to utilize existing astrophysical foundational models for efficient and scalable astronomical surveys.**

**Rating:** 5
**Confidence:** 2

**Review:**

## Summary:
This paper presents a scalable, unsupervised pipeline for identifying rare astrophysical objects in the joint Euclid-DESI dataset by leveraging existing foundation models: AstroPT, AstroCLIP, and AION.
* To better utilize the foundational models as an ensemble by addressing the misalignment between foundational models, the work utilizes density estimation in embedding spaces and a multimodal misalignment score to isolate physically meaningful anomalies from instrumental artifacts.
* The work demonstrated "representation-relative" nature of anomalies of foundational models. The result validated the intrinsic bias in outliers representation of each foundational model due to different model architectures and training objectives prioritization.

## Strengths:
* The main contribution is the establishment of a robust methodology for using foundation model embeddings as a "natural substrate" for scientific discovery at survey scale. The method gives a line-of-sight to utilize multiple foundational models as an ensemble for generic astronomical survey model especially with multimodal dataset.
* The systematic cross-model ranking-transfer analysis provides significant insight into how inductive biases affect what a model deems "unusual," shedding a direction for further improving foundation models for astrophysics. The two-stage workflow—combining density-based detection with similarity search—offers a practical and scalable approach for future large-scale astronomical surveys.

## Weaknesses:
* A primary concern is the lack of sufficient data for rigorous verification. As the authors acknowledge, the evaluation relies on a qualitative case-based inspection of a small number of candidates and proxy predictive tasks. Without a labeled "ground truth" for anomalies, it is difficult to quantitatively measure the pipeline's true discovery performance or its false positive rate beyond the qualitative filtering of artifacts. This paper would clearly meet the ICLR standard if some quantitative can be presented to demonstrate the performance of this two-stage discovery workflow can actually perform at a discovery task.
* The work provides limited major innovation in tackling the core machine learning problem of unsupervised astrophysical survey. The work primarily benchmarks existing models with "lightweight adaptations" instead of a proposing new methodology to better utilize existing foundational models (e.g. derive an ensemble solution) to further improving foundational models (e.g. with fine tuning). While what has shown here demonstrates strong empirical value, the work would bring in high value if it can further condense the pipeline into a robust solution based on existing foundational models for survey tasks that can mitigate the intrinsic biases. The two-stage workflow qualitatively shows the potential of such solution but it can be further supported with quantitative results.

Recommendation: borderline reject for the AI4S workshop. While the methodological novelty is modest, the insights regarding representation-relative anomalies, the pipeline to utilize existing foundational model to conduct effective discovery while reducing intrinsic biases in the embeddings and the demonstration of the power of multimodal input data bring inspiration in the field to further pursue a generic astrophysics foundation model. Would be above acceptance threshold if the author can provide quantitative results to show the effectiveness of this 2-stage discovery workflow.

---

### Official Review · Reviewer_peCU · 2026-02-25
**A nice analysis of embeddings from astrophysics FMs on a new dataset that allow for more careful diagnostics**

**Rating:** 5
**Confidence:** 4

**Review:**

The paper adapts three different astrophysics foundation models to a new Euclid-DESI dataset, extracts embeddings from the models in inference, and then fits a density estimator to the embeddings to analyze the anomalies. The authors then show differences and commonalities between the different models in predicting these outliers and explain how training objectives can change how these anomalies are represented.

Strengths:
- The paper is a nice analysis on what different FMs "learn" and the authors expose this through understanding how they represent anomalies
- The paper's motivation is strong and the application area is of general interest to the SciML community
- Some interpretability analyses are quite interesting and relevant to practitioners in Astrophysics, who aim to align such FMs to various downstream tasks. Diagnostics are critical to evaluating the capabilities of FMs (similar to LLMs) and this paper contributes in this direction

Weakness:
- The paper's writing could be improved - without too much astrophysics background, I found it difficult to place the results in the appropriate context. The methods could be explained more, with more informative figures (and captions) that tell the reader what the data is, how the physics (redshift etc.) is quantified, how each of the three FMs capture this information - there is some minimal text, but here informative diagrams showing the input, the processing (token shapes etc.) and what is extracted could have been really helpful.
- Each FM is adapted different (one is re-trained, one is finetuned, one is change with a different encoder). This confounds the analysis and it is unclear where these choices come from. This relates to the previous point, where more space could have been used for this purpose. Other details like train/val/test splits etc. should be in the main text.
- The paper's focus could be improved - either on the FM adaptation side (effect of zero-shot, few-shot finetuning on downstream performance), or the physical interpretation side - other estimation strategies like KNN for more interpretability (section 6.5 is nice but very short) or more exploration of the numbers in Tab. 1 (I'm unsure how to interpret a number like 21.6 retention physically). Further, without ground truth anomalies, it might be more useful to go deeper on the interpretation aspect. Another example is Fig 2.- a key ablation would be training the same backbone with different objectives to verify this behavior (it does not have to be SOTA as the FMs, but would provide insight into why we are seeing this behavior)

Overall, the paper presents a practical diagnosis on FM models but lacks sufficient details and interpretability of the results.

---

### Official Review · Reviewer_Xw5p · 2026-02-25
**Benchmarking FMs for anomaly detection in astrophysical datasets**

**Rating:** 7
**Confidence:** 3

**Review:**

This paper benchmarks three pretrained FMs and evaluate their performance on anomaly detection in astronomical data. The authors use two datasets (Euclid) and DESI (spectroscopy) and create a subset with matching modality pairs. They then investigate how much different pretrained models agree on detecting anomaly and evaluate learned represenations based on unimodality/multimodality.

This work is well studied. As they note, such study is highly needed for the planned large scale surveys (Euclid). The methods for evaluating models' performance are well-chosen and justified. The authors highlight multiple findings in the paper that are helpful for follow-up works and possibly other scientific domains e.g.:
- Anomalies are highly dependent on how representations are learned.
- Multimodal representation learning helps as a filter against artifacts in unimodal representations

The authors use cross-model ranking transfer with retention at top 1% and top-10% anomaly sets to show overlap and highlight meaningful re-orderings among different models. This is also a good method to show how representations are learned based on different training objectives.

The paper is structured quite well. Comparing anaomalies based on learned representations is a new/novel approach as opposed to the universal anomaly score evaluations.

Pros:
- extensive benchmarking, comparing different architectures
- using existing datasets (Euclid and DESI) to create matching pairs to study and evaluate
- useful metrics and methods for evaluation of the results, e.g., cross-model transfer

Cons:
- limitations to adapt. The authors mention they had to use a U-Net to adapt the resolution for the AION model and admit that spatial information is sacrificed in the process
- Limited validation: without ground-truth anomaly labels, evaluation is dependent on qualitative results and proxy tasks. e.g., the authors correctly show that unimodal models are prone to retain artifacts, multimodal fusion can be used to filter these artifacts. But without ground truth, it's difficult to know whether multimodal learning helps advance the field or it just shows which artifacts we see.

Some suggestions to improve:
- Controlled source-injection tests (the authors mention this as future work). Inject synthetic lenses/AGN-like spectral perturbations and measure detection/retention under ranking transfer
-  Compare Masked autoregressive flow (MAF) to classic baselines like k-nearest neighbour (kNN), Isolation Forest, and Gaussan Mixture Model (GMM) and check rankings and overlaps at top-k
- A small subset of top-N anomalies labeled by experts would be great and could strengthen the findings of the paper
- The authors used an adapter for AION. It would be nice to try and show rankings without the adapter to see how the adapter impacted the results.

---

### Meta-Review · Area_Chair_6ob5 · 2026-02-27

**Recommendation:** Accept (Poster)
**Confidence:** 4

**Metareview:**

This submission has received five reviews. Two very positive reviews with a "strong accept" and one "accept". The remaining three reviewers have rated the paper with a "marginally below acceptance threshold".

After reading the reviews, I recommend this paper for "acceptance" and ask the authors to consider implementing the feedback given by all reviewers into the camera-ready version of the paper.

---

### Decision · Program_Chairs · 2026-03-03

Accept (Poster)